# Modular Electromagnetic Transducer for Optimized Energy Transfer via Electric and/or Magnetic Fields

**DOI:** 10.3390/s23031291

**Published:** 2023-01-23

**Authors:** George-Claudiu Zărnescu, Lucian Pîslaru-Dănescu, Athanasios Tiliakos

**Affiliations:** 1Laboratory of Sensors/Actuators and Energy Harvesting, National Institute for Research and Development in Electrical Engineering ICPE-CA, 030138 Bucharest, Romania; 2National R&D Institute for Cryogenic and Isotopic Technologies (ICSI), 4 Uzinei Street, 240050 Râmnicu Vâlcea, Romania

**Keywords:** magnetic and electric field, transducer, wireless energy transfer, coil self-capacitance, air core RF transformers

## Abstract

In this paper, a modular electromagnetic transducer that achieves the optimal transfer of energy from the electric and/or magnetic fields is proposed. Both the magnetic field resonance coupling and the influence of the electric field near the copper transducers of the printed circuit board and inside the FR4-type epoxy material are considered. In our printed arrays of flat transducers, we consider face-to-face capacitances for the study of resonance coupling. Because the space between coil turns is almost double the plate thickness, the coplanar capacitance can be ignored for frequencies under 2 MHz. A radio frequency (RF) transmitter and transducer were built to demonstrate the increased energy transfer efficiency when using both electric and magnetic fields in the near-field region. The transversal leakage flux coupling of a long RF coil was more efficient than a simple axial magnetic field coupling when using pancake transceiver coils. The optimal configuration having one long coil at the base and two or more flat coils as capacitor plates near coil ends generated the highest tandem of magnetic and electrical fields. A power regression tool was used to convert and simplify the transducer current and voltage variation with distance. In this regard, the current change corresponded to magnetic field variation and the voltage change to the electric field variation. New formulas for estimating the near-field region and the self-capacitance of the RF transformer coil are proposed; the optimal function in the frequency domain for a given transducer distance was defined by simulation.

## 1. Introduction

Tesla’s work was very important in clarifying the non-radiative mid-range power transfer behavior of extremely low frequency (ELF) waves (6–60 Hz), where very low attenuation of waves with increasing receiver distance was observed: under 1 dB/1000 km and between 9–12 dB/1000 km for 2 to 15 kHz resonant frequencies, respectively. Tesla observed that long-range and mid-range power transfer can be achieved by electrostatic fields and electric charge variation at low frequency, leading him to envision resonant systems charged at huge potential differences (10 MV). Today, such systems are restricted (noncommercial use), due to electromagnetic compatibility (EMC) standards for electric and magnetic field (EMF) limits. Such limitations, aiming at reducing electromagnetic pollution, can also be respected by using modular and sandwiched shielding materials, such as silica/carbon nanotubes, polyimide-silver nanowires, and cellulose nanofiber composites, that can absorb electromagnetic radiation [1].

Wireless power systems operating at 100 MHz and lower frequencies can be more efficient and can transmit more power compared to microwave (RF) systems limited to 10 W m^−2^, the safety limit for human exposure, operating at 2–100 GHz. Between 50–100 MHz, even class E or D amplifiers are difficult to control and rising switching losses are a big issue when the operating voltage is higher than 100 V. Under 27–50 MHz, switching losses can be minimized by using silicon carbide (SiC) semiconductors with minimum parasitic capacitance.

From 2011 to 2019, there have been reports of resonant inductive systems for wireless power transfer at 3.8 MHz, 6.8–7.6 MHz, or 9.33 MHz, and with one to five coil resonators, with a maximum transferred power of 10–60 W and maximum usable distances of 20–100 cm [2,3]. In 2016, a combined inductive (30 × 30 cm^2^) and capacitive (61 × 61 cm^2^) system for wireless power transfer with inductive and capacitive (LC)-compensated topology at resonance was developed. It transferred 3 kW at a 1 MHz switching frequency and a 15 cm air gap with an efficiency of 94.5% [4]. Based on this, our work proposes coupling electrostatic and magnetic fields to optimize wireless power transfer by constructing a class-E fast-gate-controlled resonant Royer oscillator, to obtain resonant frequencies in the 1–10 MHz interval. A Royer circuit is a circuit with two metal-oxide-semiconductor field effect transistors (MOSFETs), with zero voltage switching (ZVS) tandem control that allows a significant reduction in switching losses. Thus, the transistors can be used at frequencies of 1–5 MHz and primary coil voltages over 120 V, without significant temperature spikes. ZVS control is realized by a TPS2814P fast-gate driver and by twin inductive “kickback”-type signals; these are produced by two ultra-fast Schottky suppressor diodes (or rectifiers) with a fast response of 50–75 ns (UF5404, UF5408), and with the help of a voltage divider designed for the maximum supply voltage of 120 V_dc_. Gate MOSFET signals are attenuated to 5–8 V by the voltage divider and used in tandem (pins 1 and 3 as input, and 5 and 7 as output on the transistor gate) by the TPS2814P circuit (see the system with a TPS2814P gate driver presented in the paper). The selected frequencies are a compromise between longitudinal wave attenuation for shorter distances (up to several meters in the near field zone) and increased wave directivity and propagation along the ground or other conducting surfaces.

In single transmitter to single transducer wireless power transfer, more precisely for flat magnetic loops with a low number of turns, there is theoretical and experimental evidence that efficiency decreases under the 50% limit, if the transmission distance is much larger when compared with both transmitter and transducer diameters. An efficiency close to 100% will be achieved if the transmission distance is comparable or maximum two times greater than the magnetic loop’s diameter [5,6].

Here, we propose a scalable method that works well at 1–10 MHz, to enlarge the total transducer surface and to improve electric and magnetic coupling; this is achieved by positioning two or more transducers near each other and by arranging multiple printed circuit boards (PCBs) and double-sided coils in succession to form resonating stacked capacitors, with Teflon sheets or FR4 (epoxy laminated fiberglass) as dielectric materials.

In our design, power transfer efficiency is maximized by: (i) choosing the dielectric sheet thickness to be under 1 mm for the secondary receiving pancake capacitors and by increasing the total surface of the receiving capacitors, and (ii) ultra-violet (UV) printing a spiral with 3–5 mm thickness and decreasing the total number of turns for a single pancake coil: 22 turns for 1.5 MHz, 11 turns for 2 MHz, and 8 turns or less for frequencies of 3.8–7 MHz [7]. All resistances from primary and secondary coils need to be decreased as much as possible [8]. Thus, we use a single- or double-faced copper layer with a thickness of 20–70 μm for frequencies of 1–10 MHz in all transducer coils, and 7 to 14 insulated stranded wires (Litz wires) with a 0.2 mm maximum diameter for transmitting coils. By using the maximum allowed surface for currents of 0.1–0.3 A, the resistances of transmitting and receiving coils can be minimized to increase the total efficiency. UV printing technology is the most suitable manufacturing process for frequencies of 1–10 MHz to minimize space usage. Below 1 MHz, we will be forced to use multiple parallel layers or strands that occupy most of the space just for a single receiving current pathway, thus rendering this process as prohibitively inefficient.

The C3 improved configuration enables separate longitudinal electric and magnetic field generation in the plate mounting direction. The C3 configuration consists of the long coil, two spiral-coil end transceiver plates for resonant capacitive and inductive topology, together with four resonant flat-coil transducer plates arranged for both capacitive and inductive coupling. These longitudinal fields enclose similar plate surfaces to avoid useless radiation. The dielectric between and inside the plates produces the resonant frequency. The 3 mm gap between the turns is almost double the thickness of FR4 (1.5 mm). This way, only the stray capacitance between flat coils counts as active and a resonance frequency of 1.5–2.0 MHz is achieved. If Teflon sheets are used instead of FR4, the resonance frequency is shifted to 3.8–4.0 MHz [9,10].

UV printing technology depends only on the available photoresist surface and the UV light scattering area. One important consideration for UV printing is to use a perfect toner black mask, with no missing dots, printed from a high-resolution image of 1200 dpi; the laser printer can print this mask on a transparent projector-slide sheet. In the case where a photoresist layer is not desired, a direct flatbed UV printer can be employed to print the UV-cured mask directly on the copper layer surface; the process is concluded by a final etching stage. For the first method, the mask must cover the copper laminate and should be pressed over by a diffuse, light-scattering glass. In this way, the UV light from the lamp will have a uniform distribution and will be on for about 7–8 min. After this stage, the laminated copper plate with the photoresist is bathed in a 7–8% caustic soda solution for 5 min or until the cured positive photoresist is completely etched and only the photoresist mask remains (as it was protected by the first toner mask). The last UV printing stage is the usual ferric chloride etching process for removing copper.

Apart from maximizing the parallel capacitance and minimizing the internal resistances of inductors to obtain an increased coil quality factor *Q* and power (according to Miller theory [11,12,13]), at MHz or higher frequencies inductors have a self-capacitance distributed in three equal series parts; parallel inter-turn capacitance is not neglected but accumulated using this equivalence. This is similar to a constant low-frequency internal capacitance distributed between three turns. This equivalence can be explained by considering the self-resonance frequency of three times less inductance and a three times higher self-capacitance. In this case, any inductance coil with this distributed capacity is closely equivalent at any low frequency or D.C. inductance. Inner coil capacitance can be comparable to the other tuning parallel capacitances at MHz frequencies; thus, any resonant frequency mismatch due to improper coil self-capacitance considerations can lead to inefficient wireless power transfer and a lower secondary voltage. Our new inductor self-capacitance formula tends to closely follow the Medhurst interpolated data, and also has a correction factor extracted from the long ellipsoidal shape of our three-turn inductor model. This factor varies from 1 to 1.27, implying that Medhurst data may underestimate or overestimate coil self-capacitance by a 22% at maximum.

## 2. Materials and Methods

### 2.1. Methods of Design and Manufacturing RF Transducers

Few papers have made a link between the distance from the transmitter to receiver, frequency or electromagnetic field decay in the near-field region. A simplified electric and magnetic field decay formula with distance and frequency, helps to determine the upper limit where the wireless transducer retains its efficiency. As far as we know, past experiments at 500 kHz and 117 W delivered power across 1 m distances, with a 65% coil-to-coil system efficiency [14]. By using two large (over 1 m) Tesla coils, Leyh and Kennan were able to transmit most of a 1000 W power supply mainly by capacitive coupling at a four meter distance with 30–35% efficiency; the experimental setup was similar to Tesla’s work, with one conductor as the earth and a non-ionized capacitive path (some Tesla papers suggest an ionized path at a 60 kHz oscillation frequency) [15]. At that frequency there is little radiated power at 4 m compared to the 5 km wavelength. Still, that system is very bulky at that frequency and has little commercial applicability. On the other hand, if we use a resonating magnetic field at 10 MHz and copper coils of 60 cm diameter, the wavelength becomes 30 m and some power is lost into space as radiation.

A magnetic loop or dipole can be considered infinitesimal only if its radius is less than λ/6π (even reaching as low as λ/10π), where *λ* is the wavelength, in order to obtain a near-constant current distribution. This is the limit where the far field starts, thus in the near-field region the coil radius must be higher or equal than this limit, which comes from the approximation of the 1st-order Bessel function expanded in series. To obtain this, we must determine the general solution of a constant current loop with *n* turns. For an infinitesimal electric dipole that is omnidirectional, the distance limit is 0.12 *λ*. From these theoretical considerations, a wireless system operating in the near-field region remains efficient, at least by 50%, if its frequency is set under 15 MHz and a multi-array system is used to increase the effective aperture. The maximum mounting distance (transmitters to transducers) should be limited to 3 m [16,17]. At frequencies over 24 MHz, the maximum mounting distance should be kept under 1 m.

Besic et al. fabricated a transducer on a PCB board that transforms the electric field into a spatial displacement. The transducer comprised a stationary and a moving part—the latter is connected to the stationary part at two sides, enabling an out-of-plane rotary displacement. When alternating charges reach the gold areas, due to the electric field *E*, the transducer will start oscillating, moving the laser mirror of the interferometer system [18].

Overhauser sensors are sensitive to any orientation change, due to their solenoid induction coil. However, the experimental results from [19] demonstrated that the sensor displayed high-accuracy measurement capabilities and the orientation sensitivity was suppressed. A short-length solenoid coil was designed for omnidirectional measurements, and the magnetic polarization field was confined to an angle of only 30% parallel to the axis, ensuring that the sensor can detect a signal at any angle. An ultralow-noise primary amplifier based on an LC resonant circuit was mounted to the sensor output to increase the signal-to-noise ratio by a factor of 2.

In the work of Toney et al. [20], an integrated opto-electric field sensor was able to detect near-surface electric fields from 20 to 30 kV m^−1^, from low audio frequencies up to very high frequencies (VHF). A spatially resolved electric field measurement above the surface of an RF stripline was demonstrated. The method from [21] employs a feedback coil placed near the receiver in the sensor, to partially cancel the errors introduced by the magnetic properties of the soil. A soil sensitivity metric was introduced to quantify the effects of the soil, and this metric was used to optimize a circuit for driving the feedback coil. A similar method, presented in [21], can be used for a flat coil transducer array to detect the magnetic fields. A PCB-printed feedback coil can be mounted in front of each flat coil transducer to quantify the effects of the surroundings.

The foundation of our study starts with a design algorithm for calculating air RF transformers and RF coils (Section 2.2). The self-resonance frequency is the most important parameter to be considered when designing resonant wireless power transfer systems. When two magnetically coupled coils are slightly out of resonance, the whole system efficiency decays rapidly—a good self-resonance frequency estimation leads to an optimized design. The self-resonance frequency is directly dependent on coil self-capacitance. Although coil inductance can be calculated precisely (at the order of 0.1–0.01%) by using the Lundin formula or the Nagaoka correction factor, the self-capacitance of a coil remains difficult to estimate precisely. Many formulas have been proposed, but each is inaccurate for different geometries. Here, we propose a new self-capacitance formula based on lumped elements theory and the Miller lumped capacitor model. Two self-capacitance formulas are proposed depending on coil geometry and conductor insulation (Section 2.3).

Another purpose of this study was to analyze the behavior of electric and magnetic fields in the near-field region by measuring the voltage and current of the transducer array as a function of distance (Section 3). Electric and magnetic field amplitudes were determined indirectly by using simplified power regression formulas for both measured current and voltage. The power regression method was also applied for the theoretical near-field 3rd-order equations for a short electrical dipole (Section 2.4). These mathematical expressions are very complicated when applied to the near-field region, but can be simplified by calculating a single irrational power degree.

New theoretical transducer positioning distances and near-field limits were established as functions of frequency and directivity of aperture. These limits should be considered to optimize the efficiency of a wireless energy transfer system by choosing the suitable resonating frequency, number of transducers, and transducer coil diameter or aperture (Section 2.4).

### 2.2. Resonant Air Transformer Design

The RF transformer comprises a short primary coil made of nine turns, a long 150 mm resonant secondary air coil (or Tesla coil) with 230 turns, and two pancake coils that are connected as capacitor plates at both secondary ends. The transducer comprises four or more pancake coils, also in dipole arrangement, along the secondary coil. The magnetic coupling is made by using the transverse leakage magnetic flux along the coil and not by using the axial field. The path of the electrical field is between the four pancake coils arranged as a dipole. The transducer is printed on both sides of a FR4 PCB plate. This plate serves as the resonance self-capacitance dielectric at 1.53 MHz, and because two pancake coils are printed on both sides, the middle connection between the coils is left out until the self-capacitance is determined. In this way, at least four transducers can be connected in parallel, back-to-back, at a 4 mm or less separation distance. Receiver or transducer coils are positioned one behind another and cover the whole length of the long RF coil. Tesla coil ends are positioned in the center of the transducers. By using separate diode bridges, we supply the load with much more power, up to 25 W, at a distance of 40 mm. Additionally, the rectifying bridge diodes play an important role when dealing with both the electric and magnetic fields. Some Schottky diodes can harvest more power from the magnetic field, less voltage and more current (1N5817 rectifying diode), and some can harvest more power from the electric field, higher voltage and less current (SB1100 rectifying diode). Because our Royer zero voltage switching (ZVS) power supply oscillator was primarily designed for plasma air or vacuum tube discharge, a high voltage (1000–1200 V) and less current for the RF coil (0.2–0.4 A) was considered. Due to 230 coil turns laying on a polypropylene (PP) tube of 32 mm in diameter, significant RF magnetic and electric fields are obtained over a large area. For this RF transformer dimensions, a 1.53 MHz resonating frequency is optimal. If we want to increase the frequency to 3.75 MHz, half of this coil length (70–80 mm) and around 105 turns must be used.

Verified by antenna experiments in the far and near fields, the directivity, covered area, and distance increase proportionally to the frequency. Of course, as we further increase the frequency, the transferred power limit (smaller element surface) will decay for an individual transducer element, forcing us to add more transducer elements to gain more power. In the near-field region directivity is low; therefore, the electric length for antennas or other elements becomes much shorter than the wavelength, thus also forcing us to increase the number of additional receiver or transducer elements to enlarge the surface or directivity.

The first step to establish the resonant frequency of the RF coil transformer is to calculate the inductance and self-capacitance of the primary and secondary coils. When we design an RF Tesla coil, we consider that the magnetic coupling coefficient (*K_m_*) is weak: under 0.5 (in most cases reaching 0.2). If we want to increase the secondary voltage, the obvious option is to increase the ratio between secondary and primary inductance until we reach the desired frequency. As a secondary option, the parallel circuit quality factor should be slightly adjusted to increase the current and to reach the desired resonant frequency. For optimum energy transfer efficiency, both options, i.e., good magnetic and electric coupling, must be considered.

An ideal Tesla coil has *K_m_* = 1 and the ratio between voltages *U* depends only on the inductance *L* and capacitance *C* ratios:(1)$$\frac{U_{\sec}}{U_{pr}}=K_{m}\sqrt{\frac{L_{\sec}}{L_{pr}}}=K_{m}\sqrt{\frac{C_{pr}}{C_{\sec}}}$$

In our case, we have improved the magnetic-coupling coefficient to 0.5 by minimizing the distance between the primary and secondary coils to 2.8–3 mm. The 40 mm PP tube thickness is 1.8 mm, and we have an air gap of about 1 mm at the secondary windings mounted on a 32 mm PP tube. The primary coil is excited by a gate-controlled Royer circuit in less than 100 ns.

The number of turns for the primary coil was adjusted to 8–10 to approach the secondary resonance frequency of 1.53 MHz. If we reduce the number of turns from nine to five turns, the solenoid inductance will decrease from 9 μH down to 4 μH. Inductance is changing around 1 µH per turn in the primary. Primary coil wires are standard ones, with a 1.5 mm^2^ cross-section and polyvinyl chloride (PVC) or polytetrafluoroethylene (PTFE) insulation. The wire can withstand an amperage of 1–3 A without overheating at 1.53 MHz and 1.9 MHz (five turns). If we want to decrease all dielectric losses, the ideal insulator is PTFE, which can be used for windings and PCB plates.

For 1.53 MHz, the secondary coil inductance is near 0.3 mH and it has a significant internal capacitance of 18 pF (*C*_sec,*C*_). To reach 1.9 MHz, the secondary inductance was decreased to 0.25 mH and self-capacitance to 14–15 pF. From these experimental measurements and trial and error tests to achieve the desired resonance frequency, we observed that self-capacitance increased with the higher number of turns. This led us to the conclusion that a coefficient must be introduced in the internal capacitance formula or at least a part of this capacitance must be regarded in parallel.

The secondary number of turns was 230 for 1.53 MHz and 210 for 1.9 MHz. Secondary coil inner capacitance will add over the selected standardized 4.7–22 pF ceramic or mica class 1 capacitors for 1–7 MHz parallel resonance (*C*_2_):(2)$$C_{\sec}=C_{\sec,C}+C_{2}$$

For a precise adjustment of 1.86 MHz resonant frequency and a peak 1500 V secondary voltage, three 4.7 pF C0G (class I material with 0 temperature drift) ceramic capacitors were mounted in parallel on a PCB plate to have a 14–15 pF summed capacitance (*C*_2_). The PCB plate that was used had U-shaped aluminum cooling profiles. At 1.86 MHz, the total resonant secondary capacitance is 29–30 pF (14 pF self-capacitance); at 1.53 MHz, we have a higher number of turns (230) and the internal coil capacitance increases to 18 pF, for a 34 pF summed resonant capacitance (*C*_sec_). For further resonance frequency adjustment, we prefer to change the primary number of turns (±1 or 2 turns) and to cut the connection wires to the desired length until we reach exactly the same secondary resonance frequency value. It is the easiest way because we use a lower wire length (1–2 m) and fewer turns (5–10 turns):(3)$$\begin{array}{c} L_{\sec}=\frac{\mu_{0}\pi D_{c}^{2}N_{s}^{2}}{4H_{c}}\left(\frac{1+0.383901u^{2}+0.017108u^{4}}{1+0.258952u^{2}}-\frac{4u}{3\pi }\right)\\ D_{c}\leq H_{c} \end{array}\}$$

The secondary coil inductance is precisely estimated, 0.1% to 0.01% precision, depending on the proximity effect considerations, by using the Lundin handbook formula [22] that is very close to the Nagaoka correction (10^−5^ digits error). The Nagaoka coefficient decreases with the coil shape factor, $u=D_{c}/H_{c}$. This signifies the increase in leakage flux as the coil becomes shorter: coil length or height *H_c_* becomes smaller when compared to the coil diameter *D_c_*. As the secondary coil becomes longer than its diameter, the dispersion flux is minimized and the magnetic-coupling efficiency is increased.

### 2.3. New Self-Capacitance and Secondary Coil 3D Electrical Field Model

An accurate formula of the inner capacitance of a single layer RF coil, derived from Medhurst data and rearranged by Knight [23], can be expressed as:(4)$$C_{secC}=\frac{4\varepsilon_{0}\varepsilon_{m}}{\pi }H_{c}\left(\frac{1}{\varepsilon_{m}}+0.7096\;u+2.395{\;u}^{3/2}\right)\;,\;\;\;\;u=\frac{D_{c}}{H_{c}},\;\;\varepsilon_{m}=1\ldots 2,$$
where $\varepsilon_{m}=\left(1+\varepsilon_{r}\right)/2$ is the average permittivity factor, with $\varepsilon_{m}=1$ for the number of coils surrounded only by air; and the relative insulation permittivity is *ε**_r_*.

It is known that, because of the proximity and skin effects, the conductive effective surface decreases with increasing frequency. As the RF frequency is increased above 1 MHz, the chosen enameled copper wire diameter should be under 0.3 mm, 0.2 mm optimum, because the current sheet thickness or skin depth will be around 60–70 μm. From various standards we saw that a maximum current of 25–30 mA would pass through a 0.2 mm wire (at 1.5 MHz–2 MHz) with no heating problems. For seven Litz wire strands, 0.2 mm diameter twisted wires, a secondary current of 0.3 A up to 0.5 A can be supported, with no additional heating problems. For a simplified calculation of self-capacitance, a Litz wire will be equivalent to a bigger toroidal turn of 0.7–1 mm in diameter.

Although we can abide by the work of Massarini and Kazimierczuk (GKMR) on the self-capacitance of multi- and single-layer coils [24,25], here we propose a different approach to single-layer coils: a single turn is similar to a toroid (3D view) charge with the same polarity in all directions, and the electric field encloses the two neighboring turns of opposite polarity. To apply this model, the entire surface of the toroid is considered for three turns, or half of the surface is considered for two turns. To avoid any complications, we have rearranged all formulas in diameter ratios.

Figure 1 presents a diagram of the enclosing electric field for longer or shorter coils. *L_sec_* is the inductance of the secondary air core transformer, *C*_*sec*,*C*_ the inner capacitance of the parallel lumped element of the coil, *C_t_* the coil series self-capacitance, *N* = *N_s_* the number of turns and *D_i_* = *d_i_* is the insulated winding diameter. Assuming the Miller model for lumped capacitor and lumped element theory [11,26], an equivalent parallel capacitance of three times the coil self-capacitance should cause a coil to resonate at half of its self-resonance frequency. If we take only one third of the coil (*L_sec_*/3) and make it resonate with the previous equivalent parallel capacitance *C_parallel_*, we will obtain the same self-resonance frequency, but now the self-capacitance *C_t_* = 3 *C*_*sec*,*C*_ = *C_parallel_*. So, from this equivalence we can see that all turn capacitances are now added in parallel for one third of the coil (*L_sec_*/3). Of course, this assumption can only be true if we consider a uniform distribution of inductance and inner capacitance, i.e., all these three parts are equal. The formula below is applied when we have two different dielectrics (mediums), insulation (*i*) and air (*a*) between the circular conductors (turns):(5)$$C_{t}=\frac{C_{i}C_{a}}{C_{i}+C_{a}},\;if\;C_{i}\gg C_{a}\;C_{t}\approx C_{a}$$

*D_B_* = *D_c_* is the inner coil diameter, not to be confused with *d_c_*, the winding conductor diameter, considered without insulation. *C_a_* is air capacitance between three turns, *C_i_* is the insulation capacitance of one circular wire, *ε_r_* is the relative insulation permittivity and *ε_0_* is the vacuum permittivity.

For a toroidal turn, the length of the toroid is $\pi \left(D_{B}+d_{i}\right)$, where *D_B_* is the inner diameter of the coil, *d_i_* is the conductor insulation diameter, and *g* is the insulation thickness (2*g* is the space between turns when insulation is not considered). All the turns of the coil are wrapped tightly around each other, so the insulation diameter is the sum between conductor’s diameter *d_c_* and the 2*g* space between turns. The insulation capacitance of one circular wire can be expressed as:(6)$$C_{i}=\frac{2\pi^{2}\left(D_{B}+d_{i}\right)\varepsilon_{r}\varepsilon_{0}}{\ln\left(d_{i}/d_{c}\right)}$$

The air capacitance between three turns for our toroid model can be calculated by:(7)$$\int_{2g}^{d_{i}}\frac{d\left(d_{x}\right)}{\pi^{2}\varepsilon_{0}\left(D_{B}+d_{x}\right)d_{x}}=\frac{1}{\pi^{2}\varepsilon_{0}D_{B}}{\left(\ln\left|\frac{d_{x}+D_{B}}{D_{B}}\right|-\ln\left|\frac{d_{x}+D_{B}}{D_{B}}-1\right|\right)|}_{2g}^{d_{i}}$$
(8)$$\frac{1}{C_{a}}=\frac{\int Ed\left(d_{x}\right)}{\varepsilon_{0}\int E\Delta S}=\int_{2g}^{d_{i}}\frac{d\left(d_{x}\right)}{\pi^{2}\varepsilon_{0}\left(D_{B}+d_{x}\right)d_{x}}=\frac{1}{\pi^{2}\varepsilon_{0}D_{B}}\left[\ln\left(\frac{d_{i}+D_{B}}{D_{B}}\right)-\ln\left(\frac{d_{i}}{D_{B}}\right)-\ln\left(\frac{2g+D_{\prime B}}{2g}\right)\right]$$
(9)$$\Delta S=\pi^{2}\left(D_{B}+d_{x}\right)d_{x}=f\left(d_{x}\right)$$

However, the coil diameter *D_B_* is much bigger than the conductor insulation thickness *g*, *D_B_* >> 2*g*. So, the final logarithmic terms can be eliminated from Equation (8), to give:(10)$$\frac{1}{C_{a}}=\frac{1}{\pi^{2}\varepsilon_{0}D_{B}}\left[\ln\left(\frac{d_{i}+D_{B}}{D_{B}}\right)-\ln\left(1+\frac{d_{c}}{2g}\right)\right]$$

Then, from Equation (5), we can calculate one third of the inner capacitance:(11)$$\frac{1}{C_{t}}=\frac{1}{\pi^{2}\varepsilon_{0}D_{B}}\left[\ln\left(\frac{d_{i}+D_{B}}{D_{B}}\right)+\frac{1}{2\varepsilon_{r}}\ln\left(\frac{d_{i}}{d_{c}}\right)\right]$$

If the insulation thickness *g* << *d_c_*, then this capacitance contribution can be neglected, and Equation (11) becomes:(12)$$\frac{1}{C_{t}}=\frac{1}{\pi^{2}\varepsilon_{0}D_{B}}\ln\left(\frac{d_{i}+D_{B}}{D_{B}}\right)p$$

Note that $H_{c}=N_{s}d_{i}$, but $N_{s}=1$ for a single lumped element made of three turns:(13)$$\frac{d_{i}+D_{B}}{D_{B}}=\frac{1}{u}+1=\frac{u+1}{u}$$

It will be a complicated and unnecessary task to introduce the influence of this logarithm and the *u* ratio inside the *p* or *m* power factor formula, because we know that we can simplify this problem by considering the maximum value of this ratio to be 0.36788, see Approximation (14). From calculations, the *u* ratio contribution should not be attributed to the power factor *p* or *m*, because in this case the results will be erroneous.
(14)$$\frac{\ln{\left[\left(d_{i}+D_{B}\right)/D_{B}\right]}_{\max}}{\left(d_{i}+D_{B}\right)/D_{B}}\approx \frac{1}{e}$$
(15)$$p=1+\frac{e}{2\varepsilon_{r}}\ln\left(\frac{d_{i}}{d_{c}}\right)$$

For very thin epoxy layer insulation, between 20 and 70 μm, and a conductor diameter of 0.2–1 mm, *ɛ_r_* is 2.7–4, *p* is little over 1, at maximum 1.2. For other dielectrics, such as PVC, *ɛ_r_* is 2.2–2.4; for thicker 0.7 mm insulation and a conductor over 1 mm, *p* increases up to 1.6.
(16)$$\frac{{\left(d_{i}+D_{B}\right)}^{p}}{D_{B}^{p}}={\left(\frac{1}{u}\right)}^{p}+a{\left(\frac{1}{u}\right)}^{p-1}+1={\left(\frac{1}{u}\right)}^{m}+1$$

An average *m* of 1.5 should be considered for all cases where we do not exactly know the insulation material characteristics, see Formulas (15)–(17).
(17)$$p\ln\left(\frac{d_{i}+D_{B}}{D_{B}}\right)=\ln\left[\frac{{\left(d_{i}+D_{B}\right)}^{p}}{D_{B}^{p}}\right]=\ln\left(\frac{1}{u^{m}}+1\right)$$

It will be interesting to compute *p* and *m* exactly for various insulating materials and to exactly consider the distance between turns or insulation thickness, but this is not the case here. What is important to retain is that the *m* power factor comes from the insulation material characteristics and from winding conductor round geometry. For $k_{e}=1$:(18)$$C_{\sec,C}=\frac{\pi^{2}k_{e}\varepsilon_{0}D_{B}}{3\ln\left(1+1/u^{m}\right)}$$

The surface of the toroid is Δ*S* and is a function of *d_x_*, the variable electrical field distance between two turns in air. The minimum distance covered by the electrical field is the space between turns 2*g*, even though there is no insulation and the maximum covered distance is *d_i_*, the outside conductors’ diameter plus two times the insulation thickness.

The previous model of inner capacitance, from Equation (18) and the model below, from Equations (19)–(22), was calculated for three turns or for a short coil and is equivalent to the Miller lumped capacitor model, $C_{\sec,C}=\frac{1}{3}C_{t}$. Because the coil is short and the coil diameter *D_c_* or *D_B_* is much bigger than the height *H_c_*, *u* >> *1*, the electrical field will concentrate only around this three-turn toroid. For longer coils, the *H_c_* height increases and the cumulated electric field encloses a larger elliptic torus (elliptical Gauss–Kummer series for the perimeter) with the same 3D symmetrical view. The self-capacitance, Equation (18) can be successfully applied to any coil having 0.80 < *u* < 10.

The self-capacitance, Equation (23) can be successfully applied to any coil having 0.1<u<3. When exiting this interval or when *d_i_* ≈ *d_c_*, the first self-capacitance formula should also be considered:(19)$$k_{e}=\left[1+\frac{1}{4}{\left(\frac{N_{e}-1}{N_{e}+1}\right)}^{2}+\frac{1}{64}{\left(\frac{N_{e}-1}{N_{e}+1}\right)}^{4}+\frac{1}{256}{\left(\frac{N_{e}-1}{N_{e}+1}\right)}^{6}+\ldots \right]$$
(20)$$H_{e}=H_{c}/3=\left(N_{e}+1\right)d_{i}$$
(21)$$\frac{1}{C_{a}}=\int_{2g}^{d_{i}}\frac{d\left(d_{x}\right)}{\pi^{2}\varepsilon_{0}\left(d_{b}+d_{x}\right)k_{e}\left(N_{e}+1\right)d_{i}}=\frac{1}{\pi^{2}\varepsilon_{0}k_{e}H_{e}}\left[\ln\left(d_{i}+d_{b}\right)-\ln\left(2g+d_{b}\right)\right]$$
(22)$$\Delta S=\pi^{2}\left(d_{b}+d_{x}\right)k_{e}\left(N_{e}+1\right)d_{i}=f\left(d_{x}\right)$$

In a similar manner, the air capacitance *Ca* is also considered to be in series with the insulated conductor capacitance *C_i_*, and the power factor *m* also appears inside the logarithmic term.
(23)$$C_{\sec,C}≅\frac{\pi^{2}k_{e}\varepsilon_{0}H_{c}}{3\ln\left(1+1/u^{m}\right)}$$

For very long Tesla coils, $N_{S}\gg 500$ and $k_{e}≅4/\pi$. Inside the natural logarithm, the total number of turns is *N_S_* = 1, because we consider this distribution for a single lumped element (or a single three-turn model). Additionally, from Miller transmission-line theory [11,26], with uniform distribution of inductance and self-capacitance, our parallel self-capacitance *C*_*sec*,*C*_ can be regarded as three equal capacitances mounted in series: $C_{\sec,C}=\frac{1}{3}C_{t}$.

For extreme cases where *u* << 0.1 or *u* >> 10, other *m* power coefficients should be used to further correct the formula. When $k_{e}=4/\pi$, the ellipse is almost flat, *d_i_* << *H_C_*, thus:(24)$$d_{i}=H_{C}/N_{S}\Rightarrow f_{SRF}\left(u,D_{C},d_{i}\right)=\frac{cd_{i}}{2\pi D_{C}^{2}}\frac{1}{\sqrt{\varepsilon_{m}}}G\left(u\right)$$
(25)$$G\left(u\right)=u/\sqrt{\left(\frac{1+0.383901u^{2}+0.017108u^{4}}{1+0.258952u^{2}}-\frac{4u}{3\pi }\right)\left(\frac{1}{\varepsilon_{m}}+0.7096u+2.395u^{1.5}\right)}$$

The self-resonating frequency *f_SRF_* of a coil can be rapidly estimated by using the *G*(*u*) function from Figure 2 or Equation (24).

We can see that is desirable to apply the function *K*(*u*) from Figure 3b instead of Figure 3a for *u* ratios between 0.1 and 2, and then apply the correction factor *k_e_* for longer or shorter coils. The *k_e_* factor is related to the electric field behavior for longer or shorter coils and is also related to the number of turns. In the special case of very small air RF coils, the power factor *m* is 1, because charges are very close to the coil center and *d_i_* ≈ *d_c_* since the insulation thickness is very small. For large coils, with heights and diameters larger than 1 or 2 cm, charges are far away from the center, and *m* is 1.5. Here, we can see a resemblance to the electric dipole. When dipole charges are close to the calculated reference point, the power factor is 3/2; when dipole charges are far away from the reference point, the power factor is 3.

If we compare the very precise measurements taken from the Kyocera AVX Company, AL Series Air Core RF Inductors catalogue [27,28,29], see Table 1, our formula is still better than the Medhurst approximation for self-capacitance. Still, the error variation is quite big for the presented measurements, between 1 and 12%—occasionally, this error may increase up to 25%.

For a coil possessing 6.2 turns, an inductance of 7.6 µH, a diameter *D_c_* of 156 mm, a height *H_c_* of 43 mm and a self-resonance frequency (SRF) at 18.9 MHz, a self-capacitance of 10.5 pF was calculated instead of 9.33 pF according to the results from past SRF measurements performed at Applied Scientific Instrumentation, registered as a ham radio AF7NX [30]. For a coil with 7.2 turns, 6.73 µH inductance, 123 mm in diameter, and 49 mm in height, a self-resonance frequency of 22.6 MHz was determined. For this coil we have estimated a self-capacitance of 7.306 pF, instead of 7.369 pF according to the SRF measurements. For this, the Medhurst formula for short coils is out of range (18–20 pF). Another interesting measurement involved a 17-turn coil made from a cable with almost double relative permittivity (4 instead of 2.3). For this calculation we had to modify the power factor to 1.2 instead of 1.5. This 17-turn coil had 31.1 µH inductance, 103 mm in diameter, 49 mm in height and a SRF of 12.86 MHz.

For a coil of 80 turns, an inductance of 215 µH, a diameter of 58.4 mm, a height of 73.7 mm and a SRF point at 7.2 MHz, we obtained a self-capacitance of 2.41 pF instead of 2.27 pF from the SRF measurements by Pettit, KK4VB ham radio code [23,31,32,33]. The coil self-capacitance obtained from the Medhurst formula was 2.7 pF. For enameled copper windings, the epoxy layer has a permittivity between 3 and 4. Two Tesla coils with *H_c_* = 35.2 cm, *D_c_* = 10.235 cm, *d_i_* = 0.31 mm, *n* = 1136 and *H_c_* = 48.5 cm, *D_c_* = 6.08 cm, *d_i_* = 0.31 mm, and *n* = 1515 were constructed and their self-resonance frequency (SRF) was investigated by de Miranda et al. [34]. For these Tesla coils (last rows in Table 2), the power factor is maximum (1.2–1.3) and *k_e_* is calculated by a fast Gauss–Kummer converging series. This series was considered because some experiments suggested that the self-capacitance could be 4/π, 27% higher than the predicted one. This is true only for coils with a higher number of turns.

In terms of near-field wireless charging, the transformer (transmitter) uses both magnetic and electric fields to transfer the RF power to a transducer that functions as a flat coil and a capacitor plate at the same time.

### 2.4. Updated Theoretical Near-Field Limits for Wireless Power Transmission

The theoretical foundations will be shortly presented to prove that the power regression approximation is an excellent tool for studying the behavior of electric and magnetic fields in the near-field region. We know that the mathematical expression of a changing electrical field for a short dipole is complicated in the near-field region: we have the electrostatic field influence or time-variant stationary charges contribution (*1*/*r*^3^), the reactive electric or magnetic field part (*1*/*r*^2^) (in the boundary region around the antenna both fields are still contributing separately), and we have the final radiative electric field part where electric and magnetic fields are in phase and closely interlinked (*1*/*r*) [35]. Additionally, the total electric field has two components, one component is along the radius or distance *r* and the other is an angular component *θ*:(26)$${\left|E\right|}^{2}\left(r,\theta \right)={\left|E_{r}\left(r,\theta \right)\right|}^{2}+{\left|E_{\theta }\left(r,\theta \right)\right|}^{2}={\left(\frac{Idl}{4\pi }Z_{0}K^{2}\right)}^{2}\left[\left(\frac{4}{{\left(Kr\right)}^{4}}+\frac{4}{{\left(Kr\right)}^{6}}\right)\cos^{2}\theta +\left(\frac{1}{{\left(Kr\right)}^{2}}-\frac{1}{{\left(Kr\right)}^{4}}+\frac{1}{{\left(Kr\right)}^{6}}\right)\sin^{2}\theta \right]$$

If we rearrange the above equation, we can derive a general formula that contains all the other particular cases, regardless of the value of the angle. This general formula of a simple radiating dipole, where $X={\left(Kr\right)}^{-2}$, K=2π/λ, and *Z*_0_ is the free space impedance, can be rewritten by expressing the *E* power degree in absolute values:(27)$${\left|E\right|}^{2}\left(r,\theta \right)={\left(\frac{Idl}{4\pi }Z_{0}K^{2}\right)}^{2}\left[\left(1+3\cos^{2}\theta \right)X^{3}+\left(5\cos^{2}\theta -1\right)X^{2}+\left(1-\cos^{2}\theta \right)X\right]$$

For a current loop transducer with one or more turns, Equations (A14)–(A18) from Appendix A should be used to estimate the total received power from both *E* and *H* fields, in the near-field case. The current loop transmitted power *P_rad_* is equivalent to *P_T._* In terms of irradiance, $P_{R}/A_{R}=\frac{1}{2}c\varepsilon_{0}E^{2}\left(r,\theta \right)$, the total received power from the *E* field can be expressed as a modified Friis formula for the near-field case, where $\left(P_{R}\leq P_{T},0\leq \eta \leq 1\right)$ [36]:(28)$$\frac{P_{R}}{P_{T}}=\eta =A_{R}A_{T}\frac{1}{{\left(\lambda r\right)}^{2}}\left[\left(1+3\cos^{2}\theta \right)X^{2}+\left(5\cos^{2}\theta -1\right)X+\left(1-\cos^{2}\theta \right)\right]\leq 1$$

In Appendix A, β=K=2π/λ and the distances *r* and $X_{loop}=X$ are the same as in Equation (28). The near-field Poynting vector expressed for a loop can be re-arranged as:(29)$$S_{loop}=I^{2}\frac{\mu a^{4}K^{3}\omega }{8}\frac{\sin\theta }{r^{2}}\sqrt{\frac{\cos^{2}\theta }{{\left(Kr\right)}^{2}}+\frac{\sin^{2}\theta }{4}}$$
(30)$$\frac{P_{R}}{P_{T}}=\eta =A_{loop}^{2}\frac{\sin\theta }{{\left(\lambda r\right)}^{2}}\sqrt{\frac{\cos^{2}\theta }{{\left(Kr\right)}^{2}}+\frac{\sin^{2}\theta }{4}}\leq 1.$$
(31)$$\begin{array}{c} X_{mloop}\geq \frac{4-\sin^{4}\theta }{\sin^{2}\theta \left(1-\sin^{2}\theta \right)}\frac{1}{D_{T}D_{R}}\\ \sin^{2}\theta \left(1-\sin^{2}\theta \right)X_{loop}^{3}+\frac{\sin^{4}\theta }{4}X_{loop}^{2}-\frac{16}{D_{T}^{4}}\leq 0 \end{array}\}.$$
to obtain:(32)$$\begin{array}{c} r_{mloop-1}=\frac{1}{\sqrt{8}}\frac{\lambda }{2\pi }=\frac{\lambda }{5.656\pi }\\ r_{mloop-2}=\frac{1}{\sqrt{14.928}}\frac{\lambda }{2\pi }=\frac{\lambda }{7.727\pi } \end{array}\}$$

The flat coil transducers, regarded as loops with a maximum mounting distance of *r_m_* = 8 m at 1.5 MHz and *r_m_* = 3 m at 4 MHz for the highest efficiency, can be described using the above Equation (32). For a frequency over 10 MHz, a maximum mounting distance of 1 m should be considered to obtain the maximum efficiency.

For a highest efficiency, *X_mloop_* = 8…14.928 is the minimum function value to obtain the maximum mounting distance for an omnidirectional *D_T_* = *D_R_* = 1 loop. The maximum positioning distance $r_{mloop}$ will be $r_{mloop-1}$. Formulas (29)–(32) are only applicable for single loops or short coils. This theoretical limit lies between λ/6π and λ/10π, thus this mounting distance limit should be considered for any transducer coil. For a real, short dipole $dL\ll \frac{\lambda }{4},A_{T}=\frac{\lambda^{2}D_{T}}{4\pi },A_{R}=\frac{\lambda^{2}D_{R}}{4\pi },P_{T}=R_{r}I^{2},R_{r}=20\pi^{2}{\left(\frac{dl}{\lambda }\right)}^{2}≅\frac{1}{2}Z_{0}{\left(\frac{dl}{\lambda }\right)}^{2}$, the flat coil case can also be extended as a short dipole array. From this formula, we extract the maximum usable distance *r_m_* between the transmitter and transducer, for the near-field case, depending on the coil positioning angle, frequency and directivity (surface) of the coil:(33)$$0\leq \left[\left(1+3\cos^{2}\theta \right)X^{3}+\left(5\cos^{2}\theta -1\right)X^{2}+\left(1-\cos^{2}\theta \right)X\right]\leq 4/D_{T}D_{R}$$
(34)$$E_{\theta }\to \max\Rightarrow \cos\left(\theta \right)=0\wedge E_{r}=0\Rightarrow f\left(X_{m}\right)=X_{m}^{3}-X_{m}^{2}+X_{m}-4/D_{T}D_{R}=0$$

If θ=π/2, cos(*θ*) = 0 and we obtain $X_{m}=1.743$, when $D_{T}=D_{R}=1$; this is also the maximum 3rd-order equation solution, solved by applying Cardano’s method, for an omnidirectional electric field of a very short electric dipole. For θ∈[−π,π], all cubic equation solutions are inside the $X_{m}\in \left[0.755,1.743\right]$ interval. The directivity $D_{T}\;\mathrm{and}\;D_{R}$ increases to 16 or 20, with the higher number of transducer elements, $X_{m}\in \left[0.42,1\right]$. For a single loop, a cubic equation can be extracted from Equations (30) and (31). In this case, for all angles between θ∈[−π,π], we obtain $X_{m}\in \left[4,8\right]$, with the exception of when *θ* is equal to 0, π, and −π where sin(*θ*) becomes zero and the cubic equation is undefined.

The flat coil transducers, regarded as dipoles, the maximum mounting distance *r_m_* = 17 m at 1.5 MHz and *r_m_* = 9 m at 4 MHz, for best efficiency, are described by the formula:(35)$$r_{m}=\frac{1}{\sqrt{X_{m}}}\frac{\lambda }{2\pi }\approx \left(\sim 0.0779D_{T}+0.0125\right)\lambda \approx \left(\sim 0.0779\frac{\pi^{2}}{c}D_{b}^{2}f+0.0125\frac{c}{f}\right)$$

As we can see, if *X_m_* has the minimum function value, different from 0, we should obtain the maximum mounting distance *r_m_*. Both the electric dipole and the current loop limits can be calculated by visually representing each function graph and by identifying the minimum inflexion points of *f*(*θ*,*X_m_*). All 3rd-order electrical dipole solutions are identified visually when the function *f*(*θ*,*X_m_*) = 0. After all the dipole solutions are extracted, we observe that we can further interpolate the results by using Equation (35).

When *D_R_* = *D_T_* = 1, both antennas are omnidirectional with *r_m_* = 0.12 λ. For a short dipole flat coil case *D_R_* = *D_T_* = 1.5, *r_m_* = 0.14 λ; for a two flat coil dipoles array *D_R_* = *D_T_* = 2…3, *r_m_* = 0.2 λ; for eight or more dipoles *D_R_* = *D_T_* = 12, *r_m_* ≥ λ. These limitations from Equation (35), only apply to where the near field is located. A similar approach was presented by Schantz, in experiments with antennas to prove their gain variation with distance [37]. Here, we extract the maximum recommended distance *r_m_* where the flat coil should be mounted. The gain or directivity *D_T_* is replaced by an effective coil or aperture diameter *D_b_* and frequency *f*.

Each flat coil is assimilated to a short dipole because the wire length or total turn length is much shorter than the 1.5 MHz wavelength (200 m). As we know the directivity *D_R_*, defined as the solid angle 4π/Ω_A_ (steradians), i.e., about a 2.7π solid angle for a short dipole. The average transmitted power in the near field per unit solid angle is *P_T_*/4π; as the distance from the transmitter is increasing, we need to consider increasing the number of resonating coils or the total transducer surface.
(36)$$E\left(X\right)=aX^{b_{p}+1}=aX\left[\left(1+3\cos^{2}\theta \right)X^{2}+\left(5\cos^{2}\theta -1\right)X+\left(1-\cos^{2}\theta \right)\right]$$

Regardless of the value of the *a* constant, the *b_p_* coefficient (the same when using a power regression approximation) can be determined by using the logarithmic transformation:(37)$$b_{p}=\frac{\log\left[\left(1+3\cos^{2}\theta \right)X^{2}+\left(5\cos^{2}\theta -1\right)X+\left(1-\cos^{2}\theta \right)\right]}{\log X}$$
where $X={\left(Kr\right)}^{-2}$.

By using computational algorithms and changing the angle *θ* between ±π in radians with a 0.126 radians step, (Figure 4a), increasing the distance (*r*) from 0.01 m (near the transmitter) to 2.5 m with a 5 cm step and at the same time modifying the frequency between (1–10 MHz), we observed that the electric field has two different inflection zones.

There is one zone, at certain frequency steps or inflection points, where the power degree coefficient is at a minimum of −3 (only the electrostatic field influence is present) and another inflection zone where the electrical field decreases much more rapidly, (Figure 4a) with −3 to −4, or more power degree, when the distance is over 3 m and the frequency is up to 15 MHz.

As the frequency and distance increase, (Figure 4b), a much faster attenuation of the electric field is observed in the near-field region. For frequencies between 1–10 MHz and distances from 0.5–2.5 m, the power degree is still around −3.2–−3.8 for the stationary field amplitude. For frequencies higher than 15 MHz and distances higher than 3 m, the stationary field amplitude seems to decay faster (approaching −5) approaching the Fresnel region situated at the boundary of the near-field limit and the far-field starting point.

In Figure 4c we notice that despite of the E electrical field angle variation, the maximum power degree for voltage is around 2.2, at fixed a 0.3 m distance from the transmitter or E field origin point. From this we can estimate that the electrical field amplitude changes with a 3.2 power degree.

## 3. Results

### Wireless Power Transfer Measurements

To demonstrate the increased efficiency of wireless power transfer using both the electric and magnetic fields [4,38], measurements were conducted by using four configurations (Figure 5):(i).The secondary long coil of the resonant air transformer was coupled by using the axially magnetic field with a single double-sided flat coil transducer, C0 configuration;(ii).The secondary long coil of the resonant air transformer together with one flat pancake coil as the capacitor plate was coupled by using both the magnetic and electric fields, with the same flat double-sided coil-capacitor plate mounted as the receiver or transducer, C1 configuration, with a diode bridge comprising 1N5817 rectifier diodes;(iii).The secondary long coil of the resonant air transformer was coupled by using only the magnetic field with four double-sided transducer plates mounted near and behind one another, (see Figure 5 (top)), C2 configuration, with SB1100 rectifier diodes;(iv).The secondary long coil of the resonant air transformer together with two flat pancake coils connected as capacitor plates were coupled by using both the magnetic and electric fields with the same four transducer plates with double-faced pancake coils, regarded also as the capacitors, C3 configuration, with SB1100 rectifier diodes. The flat pancake coils of the resonant air transformer have both sides connected in parallel to be used as capacitors plates, thus eliminating their internal capacitance.

The equivalent circuit of the wireless power system (Figure 5 (bottom)) comprises a DC 7809 or a 7812 voltage regulator, a gate voltage divider, a TPS2814P 40 ns fast gate driver, a Royer circuit with two STP24N60M6 MOSFET transistors and two UF5404 or UF5408 ultrafast recovery rectifier diodes in which ZVS tandem control is used, a 60 to 120 V DC voltage booster is used as a power supply for the primary coils, a 100 µH filtering ferrite, a RF air transformer, two pancake plates at the end of the air transformer secondary, four double-sided flat pancake coil transducers, four SB1100 fast rectifying diodes bridges, a 680 µF 400 V electrolytic capacitor, and the load. The ZVS control allows a significant reduction in switching losses, thus the transistors can be driven at frequencies between 1 and 5 MHz and voltages of the primary coil over 120 V without overheating. Zero voltage switching (ZVS) control is realized by the TPS2814P 40 ns fast gate driver and by each of the two inductive “kickback”-type signals. These signals are taken by two ultra-fast Schottky suppressor diodes (or rectifiers) with a response of 50–75 ns (UF5404, UF5408), and with the help of the voltage divider designed and calculated for the maximum supply voltages of 80 or 120 V. The gate MOSFET signals are attenuated to between 5–8 V by the voltage divider and are used in tandem (pins 1 and 3 as inputs, and, 5 and 7 as outputs on the transistor gate) by the TPS2814P circuit. The maximum electronic control delay time was estimated to be around 160 ns, implying that a frequency of maximum 6–7 MHz can be successfully applied to control the MOSFET transistors. The parasitic capacitance and on-off transistor switching time can be lowered further by choosing a transistor manufactured with the latest SiC technology. The measured transducer current and voltage at different distances and configurations is presented in Table 3.

The C0 starting configuration, where only the axial magnetic field is used, presented the worst results for the long coil and the pancake transducer, because the received power was low and the efficiency was under 2%. From the experiments, the magnetic coupling was stronger when we changed the flat coil position to be along the transformer coil (Figure 5 (top)). It is obvious that the magnetic field lines pass through a larger area for both the transmitter and transducer coils, even if this magnetic flux is the dispersion flux. Therefore, if in the near- or far-field region, the transmitter and transducer area should be large enough to receive more power. We have two solutions to enlarge the total surface: the first one is to build coils with larger diameters, but the number of turns will decrease with frequency, until we have only a single turn coil; the second is to have a higher number of flat coil transducers, each connected to separate RF Schottky rectifiers, but the final DC supply will be in parallel connection.

Let us analyze the C1 configuration. Because the secondary transmitter coil is electrically and magnetically coupled with only one pancake coil and the transducer is connected to a full bridge rectifier made of four 1N5817 Schottky diodes, the electromagnetic energy will be directed to the charge carriers, current will rise, and the voltage will be quickly limited by the diodes to a maximum of 35 V. Despite the large electric field detected near the secondary transformer, a voltage up to 300 V was measured when using different Schottky diodes with a high-voltage capability (UF5404, UF5408), but the measured current was low, not exceeding 10 mA. One flat transducer using fast-recovery low-voltage (1N5817) or high-voltage diodes (UF5404) collect a power with a of maximum 3 W (Figure 6a). Because our wireless device is efficient over a range between 10–100 mm, SB1100 diodes were chosen to comply with the 130 V rectifier voltage limit and with the higher measured current for one flat coil element, 40–50 mA see Table 3. It was observed from multiple measurements that a maximum power of 6 W was obtained in this case, double compared to previous cases, at a distance of 30 mm when choosing SB1100 diodes (Figure 6b).

From the experimental data extracted over a wide distance range, between 25 mm–300 mm, we have determined that the current (Figure 6a,b) and voltage (Figure 7a) always have a proper power regression coefficient that fits the data. The power order for DC current reaches two and for voltage reaches 1.5 to 2; the reported distance *x*_0_ was chosen to be 1 m, so the following expressions can be derived:(38)$$\begin{array}{c} I=I_{0}{\left(x/x_{0}\right)}^{-bp}\\ U=U_{0}{\left(x/x_{0}\right)}^{-bp} \end{array}\}$$

Table 4 presents the fitting coefficients extracted from the experimental data and from theoretical simulations presented in Figure 4 (*E_estim_*). Since the estimated electric field variation with distance is close to the proposed power regression expressions, by using Equation (38), these pancake coils can be used as electric and magnetic field transducers.

The electric field for *E_estim_* of the 3.2 order power was estimated at 1–5 MHz frequencies and a 300 mm maximum transducer positioning distance (Section 2.3). If we look at the measured voltage graph, it seems that the voltage drops faster the higher the distance (25–80 mm range), from 126 V to 50 V, when we use only the magnetic field coupling at 1.53 MHz resonance frequency (C2 configuration). When we connect two pancake plates to both secondary wire ends, we have the C3 configuration where both the magnetic and electric fields are enclosing the same area and the voltage is kept almost constant, from 127 V at 25 mm down to 122 V at 80 mm.

In addition, the current is slightly higher because of the additive effect of displacement currents from the changing electric field and the secondary magnetic field contribution. The previous DC voltage and DC current measurements were made by using four transducers and four RF Schottky diode rectifier bridges, mounted in parallel with a 680 µF/400 V electrolytic capacitor. Our experimental data confirm that by using both the resonant magnetic- and electric-coupling fields (C3 configuration), the obtained DC current, voltage and overall power is greater than the C2 configuration, where we used only the resonant magnetic field coupling. Other important manufacturing details can be observed: there are no separate resonant capacitors (Figure 7b), and the dielectric materials of all four PCB pancake plates form an internal resonant capacitor together with the flat spiral coils as capacitor plates. Considering that the standard plate thickness is 1.5 mm, the relative dielectric constant of FR4 fiberglass epoxy resin is 3.7–4, and the turn spacing or copper layer is almost double the plate thickness, only the 250–270 pF capacitance between the two copper spirals is relevant [39]. Another small internal capacitance difference of 20 pF can appear due to copper layer misalignment and turn spacing, but it does not significantly affect the resonant frequency of 1.53–1.58 MHz. The total measured inductance of the two spiral coil faces connected in series is 36 μH, so a single spiral of 11–12 turns has an inductance of 9 μH. Rectifier diodes with a very fast recovery time (below 100 ns) and a low forward voltage drop (0.3–0.6 V) plays an important role in improving the total wireless system efficiency. As the DC 12 V power supply results in an output current between 4–4.3 A, we can calculate a 51 W input power used in the installation. Because the DC booster has a 10 times conversion ratio and 120 V DC going from the booster to the RF power supply [40], the duty cycle is 0.9 and the conversion efficiency decreases to 75%, instead of 80–90%, when we used lower boosting ratios (2–6). Around 6 to 9 W are lost in power resistors used for the active control and synchronization of the power transistors gate signals.

The RF device is a modified Royer oscillator with a gate controller, that runs at a 90–120 V variable DC input voltage. From the above results, we can conclude that only 32–38 W of power are produced and transmitted by the RF power supply. If all the pancake plates with four transducer coils are placed 25 mm apart from the transmitter, they can collect up to 25 W of DC power (Figure 8b), i.e., producing a receiver transmitter efficiency of 72%. Near the 60 mm distance, the collected power drops to 10 W, and the efficiency is decreased down to 32%.

## 4. Discussion

Our parallel equivalent self-capacitance formula is derived from the lumped element and the Miller theories—recent findings about the self-resonance frequency (SRF) for coils also gives credit to the lumped element theory. The Grandi–Kazimerzuc–Massarini–Reggiani (GKMR) theory about self-capacitance, widely used in the works of Knight [23,24,25], considers all turns to be in series connection. Although this is true locally for the separation materials (i.e., air, conductor insulation and coil-formers), the electric field of the coil behaves more similar to the transmission-line theory—the coil is essentially an antenna.

The toroidal interpretation of the 3D coil spiral is another geometry to be considered when calculating the self-capacitance of round coils. This geometry produces simple and accurate approximations of the electric field behavior. The self-capacitance formula was compared to the data from at least 20 coil and SRF experiments. The results were accurate with a relative error between 1–12%. For a ratio factor closer to 1 with small coils having a conductor insulation thickness much smaller than their diameter, the diameter of the coil must be considered. The results were very compelling, with relative errors being minimized to 1–5%. For coils longer than their diameter and a large number of turns, the electric field flattens, forming an ellipse. In this case, two correction factors are needed to better estimate the self-capacitance, the *k_e_* flattening factor $\left(1\leq k_{e}\leq 4/\pi \right)$ and the power factor *m* (1≤m≤1.5). If the dielectric constant is not known, *m* should be considered as 1 for very small insulation thicknesses, or as 1.5 for usual dielectrics, such as PVC, and for thicker insulations. In the special cases of very small air RF coils, the power factor *m* is 1, because charges are very close to the coil center and *d_i_* ≈ *d_c_* as the insulation thickness is very small. For large coils, with height and diameters higher than 1 or 2 cm, electric charges is further from the center, thus *m* should be considered as 1.5.

From our experimental measurements and trials to achieve the desired resonance frequency, we noticed that self-capacitance increased with the higher number of turns. This led us to the conclusion that a coefficient must be introduced in the internal capacitance formula or the equivalent inner capacitance must be regarded in parallel.

Determining the self-capacitance of a coil and its SRF is an imperative task when designing wireless transmitters and transducers. An approximate SRF can be estimated more rapidly when using the combined *G*(*u*) function and choosing the right coil diameter.

From previous experiments (Section 3), we can see that by using both the electric and resonant magnetic coupling the total received power increases by 25%, regardless of the number of transducers. If we increase the number of transducers from one to four, the collecting area also increases by four times and the total power magnifies from 6 to 25 W. It is very important to use suitable fast rectifying diodes SB1100 instead of 1N5817, as efficiency is doubled from 2.7 to 6 W for one transducer. As frequency increases, a higher directivity is achieved, as the beam becomes narrower and the solid angle decreases (more power per distance). However, it is more difficult to obtain the same electric and magnetic field amplitudes for the same element area, thus the number of transducers needs to be increased until the desired power transfer per distance ratio is reached.

The current and voltage power regression variation with distance can be used to estimate the electrical and magnetic field amplitude. If we measure the distance with a laser or a precise mechanical system, we can choose different rectifying diodes: SB1100 voltage-sensitive Schottky diodes for electric near-field measurements, or 1N5817 current-sensitive Schottky diodes for magnetic near-field measurements; we can also design a specialized electronic circuit (comparators) to compare the estimated amplitude with the measured one, and the flat coil receivers work as transducers. We can also differentiate between the electric and magnetic fields by changing the transducer’s physical connections. When the middle connection between the face-to-face PCB inductors is disconnected, we see the voltage rise due to capacitive coupling. When we keep this middle connection active, the voltage rise due to magnetic induction is accentuated. Two pairs of transducer arrays should be used, one with the middle connection active and with current-sensitive Schottky diodes, and one with the middle connection disabled and with voltage-sensitive Schottky diodes.

New theoretical transducer mounting distances and near-field limits were established as functions of the frequency and directivity or aperture. These limits can be used to improve the efficiency of a wireless energy transfer system, by choosing the right resonating frequency and the optimum number of transducers, or by estimating the transducer active area. When directivity *D_R_* = *D_T_* = 1, both the electrical dipole transceivers are omnidirectional, at a maximum transducer mounting distance of 0.12 λ. For omnidirectional current loops, the maximum transducer positioning distance was 0.0562 λ.

From the final 3D simulation (Section 2.4), we can conclude that there are frequency and angle-dependent inflection points where the electric field power degree is at a minimum, −3, depending on the transducer distance and position (angle). These zones change in amplitude as the electric field angle moves from −π, the minimum power degree, to +π, the maximum power degree. The minimum −3 electric field power degree can be attributed to the electrostatic field contribution only. Regardless of the frequency value, there is a solution to the 3rd-order equation that maintains the power degree (or electric field decay) at the minimum value of −3. The power regression coefficients estimated theoretically are in agreement with the practical measurements (electric field decay power, −3.2), thus the theory is valid, at least over the range of 1.5–2.2 MHz.

A rough estimation of the optimal transducer positioning distance is very useful when designing a wireless system with different working frequencies.

High-voltage lines (e.g., 400 kV) produce a 50 Hz magnetic field in their immediate vicinity, which decreases with the square of the distance from the line [41]. By using the transducer presented in the work, we recovered part of the magnetic field energy in non-resonant coupling-type applications.

## 5. Conclusions

We have introduced a new self-capacitance formula for a resonating coil presenting a logarithmic growth that fits adequately to the Medhurst data, when the dielectric constant of the insulation of the coil turns is not considered and when the diameter of the conductors is considered to be much smaller than the physical dimensions of the coil. The 3D coil turn, represented as a toroid, is another geometry to be considered when calculating the self-capacitance of round coils. This geometry produced simple and accurate approximations of the electric field behavior, as well as the self-capacitance and self-resonance frequency (SRF). The self-capacitance formula was analyzed and compared against the data of at least 20 coil and SRF experiments. The results were accurate, with a relative error between 1–12%. This uncertainty also comes from the fact that the physical dimensions of the coil cannot be precisely estimated when the SRF was determined. The SRF cannot change if the physical dimensions, the number of turns, the conductors’ insulation diameter, and the insulation material are the same.

Two coefficients must be introduced into the inner capacitance formula, one to correct the Medhurst data for short or long coils and one in the logarithmic terms to account for the insulation diameter and material. The equivalent inner capacitance must be regarded in parallel and only three times in series, according to the Miller self-capacitance model.

When using both electric and resonant magnetic coupling, the total received power increases by 25%, regardless of the number of transducers. If we increase the number of transducers from one to four, the collecting area is also increased by four times and the total power rises from 6 to 25 W. If we use SB1100 diodes, more electrical field and voltage is harvested; if we use 1N5817 diodes, more magnetic field and current is harvested. Another fast-rectifying diode that can harvest both the high electric and magnetic fields could be a good compromise in order to obtain more than 6 W of power per one transducer. We need to multiply the number of transducers until we reach the desired power transfer per distance ratio and enough directivity.

To harvest more power per distance in the near-field region we have two options: either we use coils with larger diameters but we have a limited frequency range due to the physical dimensions, or we increase the number of transducers to have no certain frequency limit.

Directivity of a single transducer is still considered the same as a short dipole placed in the near-field region, even if the cross-sectional area is greater, thus we chose the second option, meaning that the directivity will surely increase with the higher number of transducers and we can still choose the desired resonating frequency.

In conclusion the second option is a more versatile and scalable method that works well for a 1–10 MHz interval. In addition, the optimal configuration of having one long coil at the base and two or more flat coils as capacitor plates near the coil ends generates the highest magnetic and electrical fields.

A power regression tool was used to convert and simplify the transducer current and voltage variation with distance. This tool can be used to estimate the electric or magnetic field amplitudes when the transducer positioning distance changes.

New theoretical transducer mounting distances and near-field limits were established as functions of the frequency and directivity or aperture for small electrical dipoles and magnetic loops. These near-field limits should be considered in the first design steps of a wireless power transfer system.

## Figures and Tables

**Figure 1 sensors-23-01291-f001:**
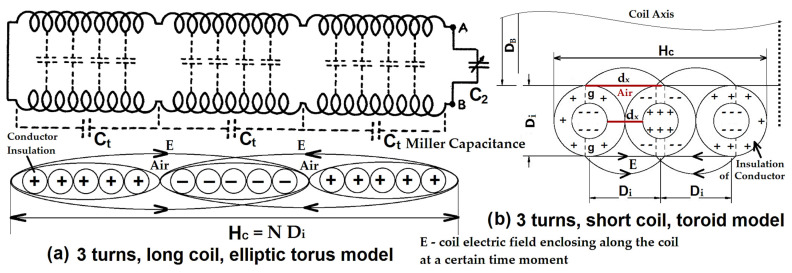
Coil antenna as a transmission line, with self-capacitance of the turns regarded in parallel; (**a**) long coil elliptic torus model with three long turns or three series *C_t_* capacitors; (**b**) short coil toroid model with three turns or three series capacitors.

**Figure 2 sensors-23-01291-f002:**
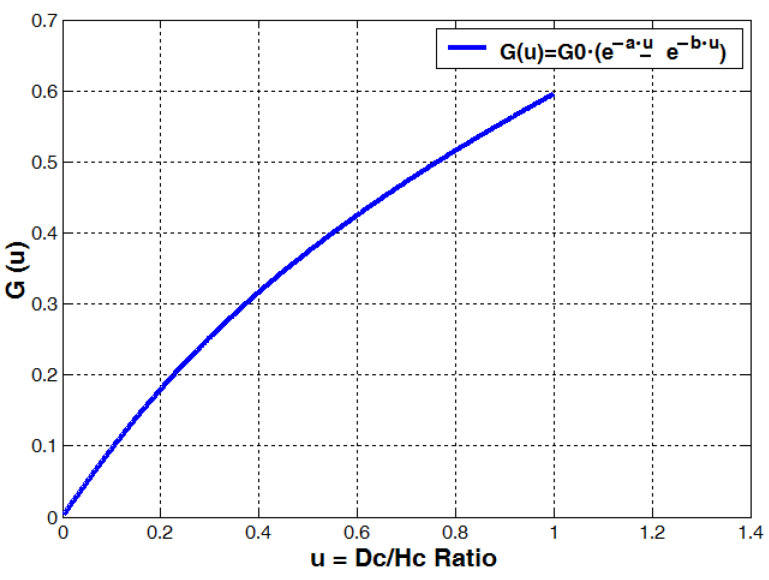
Coil geometry dimensionless function (factor) derived from the Lundin inductance formula and the Medhurst coil self-capacitance formula; here $\varepsilon_{m}=1$.

**Figure 3 sensors-23-01291-f003:**
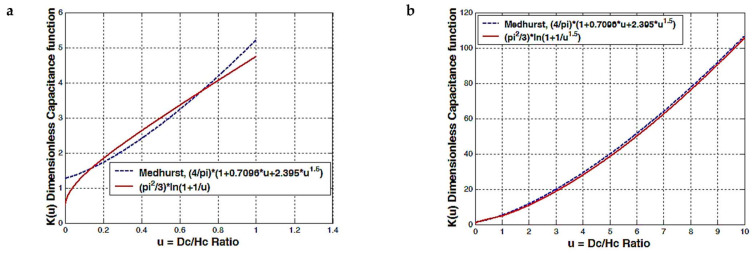
Coil capacitance dimensionless function (factor), comparison between the Medhurst formula and our theoretical logarithmic function; (**a**) 0 < *u* < 1, *m* = 1 and (**b**) *u* > 0, *m* = 1.5.

**Figure 4 sensors-23-01291-f004:**
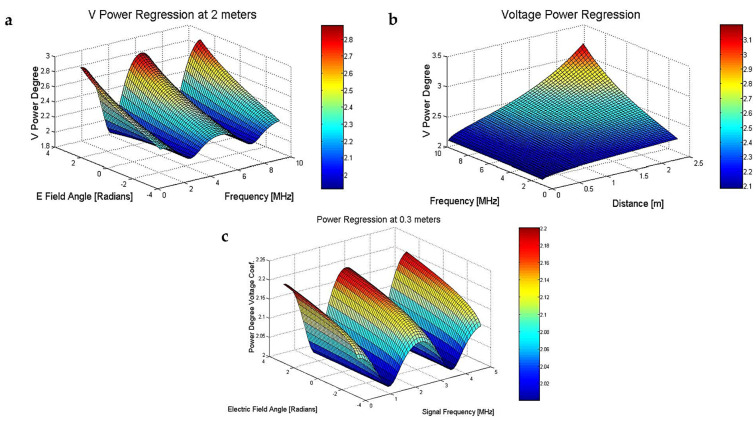
(**a**) Absolute power degree for voltage (+1 for the E field) function of the frequency spectrum (1–10 MHz) and an E field angle, in radians; (**b**) absolute power degree for voltage (+1 for the E electrical field) function of the frequency spectrum (1–10 MHz) and a transducer distance in m; (**c**) absolute power degree for voltage (+1 for the E field) at a fixed 0.3 m distance, function of the frequency spectrum (1–5 MHz) and an E field angle in radians.

**Figure 5 sensors-23-01291-f005:**
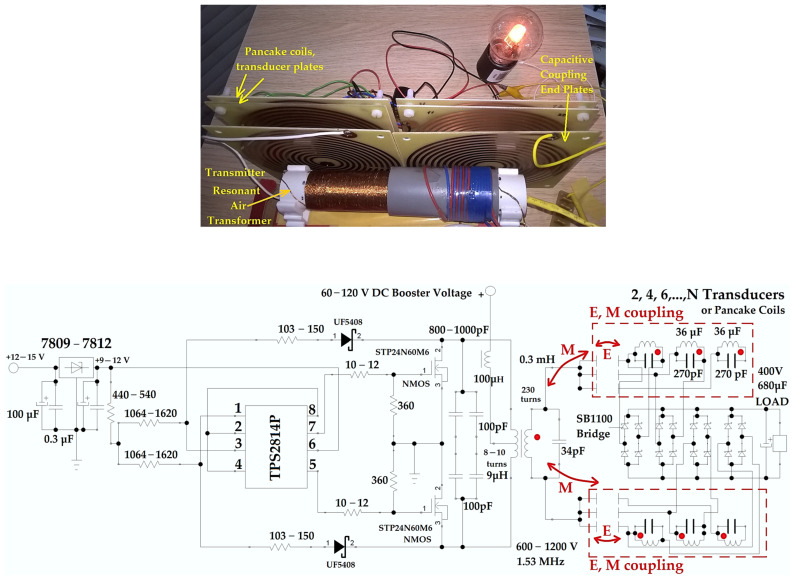
Photograph of the wireless power system for the C3 configuration, pancake coils at 30 mm distance (**top**), schematic of the system with a TPS2814P gate driver, a Royer circuit with ZVS, a resonant air transformer and two, four or more double-sided pancake coil transducers (**bottom**), where E is the coupling electric field and M is the coupling magnetic field (or mutual inductance).

**Figure 6 sensors-23-01291-f006:**
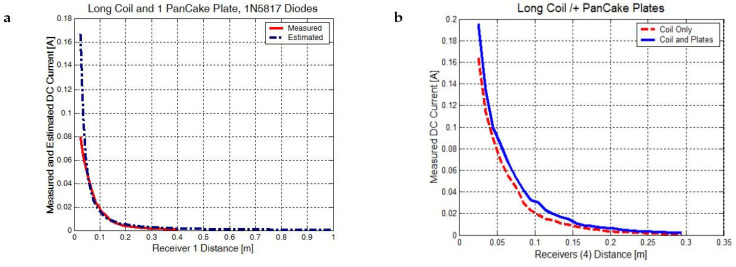
(**a**) Measured DC current for one pancake plate (C1 configuration) and a rectifier made from four 1N5817 diodes. (**b**) Measured DC current for four transducer plates, a long RF coil case (C2 configuration) or both an RF coil and pancake plates (C3 configuration).

**Figure 7 sensors-23-01291-f007:**
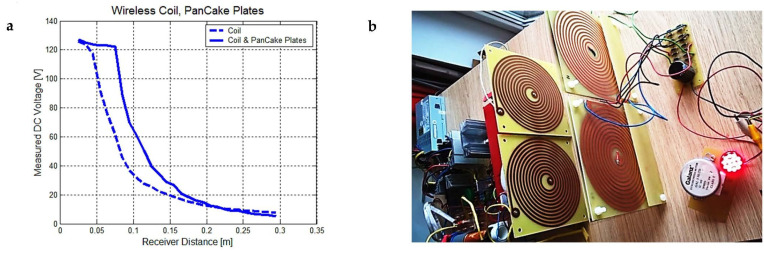
(**a**) Measured DC voltage only for a long RF coil (C2 configuration) and for both an RF coil and all pancake plates. (**b**) Transmitter and transducer plates at a 150 mm distance, producing 0.7 W to power 14 LEDs.

**Figure 8 sensors-23-01291-f008:**
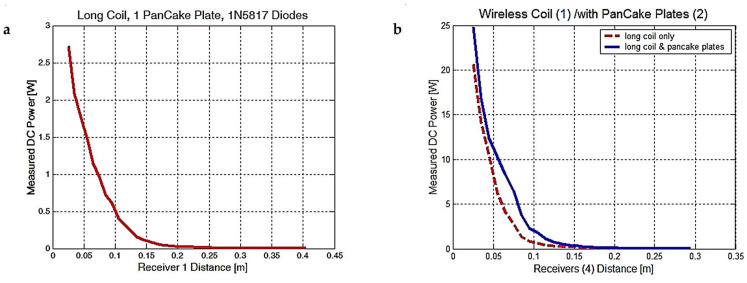
(**a**) Measured DC power for the C1 configuration, one long RF coil, one flat pancake coil as the transmitter and one pancake plate as the transducer. (**b**) Measured DC power for four transducers in the C2 and C3 wireless configurations, (C2 configuration) a long RF coil as the transmitter and (C3 configuration) both an RF coil and two pancake plates as the transmitters.

**Table 1 sensors-23-01291-t001:** RF inductors measured self-capacitance compared to the calculated Medhurst and our logarithmic formulas.

AVX RF Inductors	Part. No. AL02390N0	Part. No. AL023246N	Part. No. AL023422N	Part. No. AL12B43N0	Part. No. AL01627N0	Part. No. AL016100N
**Turns No.**	9	15	18	10	5	9
C_measSRF_ [pF]	0.2166	0.2194	0.2058	0.2610	0.1287	0.1759
C_calcMED_ [pF]	0.3026	0.3026	0.3026	0.3130	0.3514	0.3514
C_calcLOG_ [pF]	0.2205	0.2271	0.2295	0.1941	0.1636	0.1714

**Table 2 sensors-23-01291-t002:** Tesla coils and RF cable inductors measured self-capacitance compared to the calculated Medhurst and our logarithmic formulas.

SRF Tested Inductors	7.6 µH, *u* = 3.62	6.73 µH, *u* = 2.51	31.1 µH, *u* = 2.10	215 µH, *u* = 0.79	16.4 mH, *u* = 0.125	33.6 mH, *u* = 0.290
**Turns No.**	6.2	7.2	17	80	1515	1136
C_measSRF_ [pF]	9.330	7.369	4.924	2.272	6.824	6.329
C_calcMED_ [pF]	19.501	18.443	14.67	2.700	6.532	6.274
C_calcLOG_ [pF]	10.515	7.306	5.027	2.018	6.967	6.619

**Table 3 sensors-23-01291-t003:** Measured transducer current and voltage at different distances and configurations.

RF Coil and Transducers Configuration	Distance (mm)	DC Current (mA)	DC Voltage (V)	DC Power (W)
C1	25	80.0	34	2.72
C2	25	164.0	126	20.66
C3	25	195.5	127	24.83
C1	155	7.1	12.4	0.09
C2	155	8.1	18.5	0.15
C3	155	11.0	27	0.30
C1	295	1.52	3.5	0.0053
C2	295	0.8	7.63	0.0061
C3	295	2.0	5.27	0.0105

**Table 4 sensors-23-01291-t004:** Voltage and current power regression coefficients, obtained from experimental data.

SRF Transducers Tests	*I*_0_ (A)	*bp*-Power Degree	*U*_0_ (V)	*bp*-Power Degree
C1 (E+H, 1Rec)	0.000328	1.689	0.4825	1.6965
C2 (H, 4Rec)	0.000100	2.300	1.4936	1.3466
C3 (E+H, 4Rec)	0.000200	2.180	0.3740	2.2245
*E_meas_* (*H_meas_*)	-	3.180	-	3.2245
*E_estim_* (*H_estim_*)	-	3.200	-	3.2000

## Data Availability

The data presented in this study are available upon request from the corresponding author. The data are not publicly available due to institutional policies.

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
