# Peer review of "Modular Electromagnetic Transducer for Optimized Energy Transfer via Electric and/or Magnetic Fields"

_sensors, 2023, doi:10.3390/s23031291_

Round 1

Reviewer 1 Report

The manuscript “Modular Electromagnetic Transducer that Achieves the Optimal Transfer of Energy from the Electric and/or Magnetic Field" is interesting in principle, because here electric fields and induction fields are investigated and used simultaneously for energy transfer at near range. Thus, there is still no effective emission of electromagnetic waves into the distant space.

Generally for the subject and noted by the authors, there have been many preliminary works for the subject area for a very long time.

The manuscript is very extensive and contains a large amount of interspersed technical details right from the start, such as winding numbers, geometries or specific current ratings. Abbreviations and terms appear early on (e.g. line 87 "C3 Configuration"), which even an electrical engineer with a lot of previous experience with inductive and capacitive circuits cannot understand at all without introductory explanations. Readability is difficult even for an expert.

Missing from materials and methods is a simple instructive schematic for the setup ultimately presented in Fig. 5, which appears to be central to the results presented. However, there is no fundamental doubt that the presented energy transfer works somehow. The aim is probably a clever combination of inductive and capacitive effects (including resonance), but their interaction can only be guessed by me on the basis of the explanations.

A circuit diagram is mandatory and should discretely contain and represent the inductances and capacitances effective within each transducer. In addition and shown separately, the inductive coupling and (longitudinal capacitors) capacitive coupling from transmitter to receiver, which becomes weaker with increasing distance. Such a discretized equivalent circuit allows e.g. a simple simulation in free available Spice software, and makes the matter more handy and understandable.

In general, the manuscript must be structured more clearly, the multitude of interspersed and unexplained abbreviations and representations (e.g. already in unclear Fig. 1 as a basis) must be reduced and ordered. The manuscript could probably become considerably shorter and would then even have more information content for the reader. Terminology needs to be cleaner, e.g., what is a wireless coil/transformer (Fig. 5, Fig. 7). There is a lot of wire.

The central and generally interesting effect is resonance in the pancake coils, which causes electric and magnetic fields to build up and thus achieve a greater range. The shown self-capacitance formulas are interesting.

General question: If instead of several spatially parallel pancake coils only one larger coil with the same surface area would be used, wouldn't this also achieve more range? For non-radiating energy transmission, the ratio of distance (between transmitter and receiver) and cross-sectional area will always be and remain a decisive parameter. The gain of only one element with more cross-section must not be underestimated.

English needs to be improved, for this there are inexpensive or even free services in the internet.

Suggestion for the title: Modular electromagnetic transducer for optimized energy transfer via electric and/or magnetic fields

I recommend a careful and extensive revision so that even experts without very specific prior knowledge can find their way into the subject more easily.

Reviewer 2 Report

This manuscript reports a modular electromagnetic transducer that enables optimal energy transfer in electric and/or magnetic fields. New estimative formulas for the near-field region and for RF transformer coil self-capacitance were proposed. The calculations and the necessary simulations have been made to find out the optimum function in the frequency domain for chosen transducer distance.

I think the manuscript needs minor revision before acceptance for publication. The further comments are listed as follows:

1. Introduction part, if possible, some important and relative reports about EMC (Composites Science and Technology, 2023, 231: 109799) can be added to show clear background.

2. The abbreviations should be explained when they first appear in this paper, such as FR4, SiC, LC, UV, etc.

Reviewer 3 Report

Dear Authors,

Please find enclosed some comments on your article:

line 14 - an abbreviation PCB does not use in the abstract anymore. Please skip it;

lines 34, 40, 42 - please explain abbreviations ELF, EMF, EMC, and RF;

lines 35, 36, 42 - please incert a space between the number and unit;

line 48 - "these papers" - which papers?

line 53 - explain LC;

line 54 - air-gap -> air gap;

line 57 - what is Royer oscillator?

line 68, 69 - explain abbreviations PCB and FR4;

lines 71-72 - replace "UV printing" by "ultra violet (UV) printing";

line 82 - replace "ultra violet (UV)" by "UV";

line 87 - explain an abbreviation C3;

line 88 - please give a reference to "our UV printing technology";

line 112 - what is Q factor?

lines 146, 151 - what is lambda?

line 172 - explain VHF;

line 182 - replace "radio frequency (RF)" by "RF" in both cases;

line 210 - what is Tesla coil?

Intoduction is too long;

line 225 - what is 1N5817?

line 226 - what is SB1100 case?

line 229 - explain PPR;

line 244, 252 - what is Km?

Equations (1) to (35) and (A1) to (A22) - explain all the variables used;

line 702 - replace (31) to (35);

line 258 - what is PP tube?

line 260 - what is PPR tube?

line 266 - is PVC insulation allowed?

line 284 - explain C0G;

line 299 - explain u = Dc/Hc;

line 300 - what is Hc-coil;

lines 340 - 343 - explain all the variables used;

line 351 - explain D_B ans D_C;

line 415 - what is (2*g)?

line 470 - what is AVX?

line 495 - what is KK4VB?

line 854 - Where is Conclusion?

Kind regards.

Round 2

Reviewer 1 Report

The manuscript is improved, one essential aspect is still very unclear to me.

In the schematic Fig. 5 bottom the critical area of the pancake transducer coils with the inductive and capacitive couplings is not visible.  Instead only a black box with "E,H wireless power transfer" appears!

However, this is exactly where practically the entire topic of the article takes place. Relevant here: A schematic of the central function beginning from "wireless transformer" (Fig. 5 top: why is that wireless... any transformer is then wireless) up to the receiver coils, with inductive and capacitive coupling paths. This can all be represented with some capacitances and (coupled) inductances.

In addition, I personally do not at all understand the argumentation with the 50% limit and the solid angles. “A single element with bigger cross-section will not exceed the 50 % efficiency limit as described in papers [5, 6]. Indeed, a bigger cross-section means more, but still limited directivity…”

We are not yet in the really radiated far field of the relatively low-frequency arrangement (some MHz, all sizes are much smaller than a quarter wavelength). And in the near field with e.g. just inductive coupling (this strongly reduces over distance) and with idealized inductors and resonant circuits (without resistance losses) a theoretical power transmission efficiency of 100 % can be achieved over an airgap. This can be easily shown with a simulation with two weakly coupled (i.e, they are distant) inductors. Radiation losses into the far space can be neglected at sufficiently low frequencies.

And it is also clear to show and calculate that two coils of large cross-section spaced apart will always couple better than several small coils of the same total area and spacing.

In this respect, I still have certain doubts about the principal superiority of the whole concept. These doubts are only diffuse, but they will not be dispelled by a further review process.

Please address the issues, I do not want to see the manuscript again. Best wishes.

Reviewer 3 Report

Dear Authors,

after the moderate English changes this article can be accepted for a publication.
